# Policy addressing suicidality in children and young people: an international scoping review

Lynne Gilmour  , Margaret Maxwell, Edward Duncan

Nursing, Midwifery and Allied Health Professions Research Unit, University of Stirling, Stirling, UK

**Correspondence to**
Ms Lynne Gilmour;
Lyg1@stir.ac.uk

## ABSTRACT

**Objective** To map key policy documents worldwide and establish how they address the treatment and care needs of children and young people (CYP) who are suicidal.

**Design** We conducted a scoping review to systematically identify relevant key policy documents following a pre-established published protocol.

**Data sources** Four databases (CINAHL; Medline; PsycINFO; The Cochrane Database of Systematic reviews) and the websites of key government, statutory and non-statutory agencies were searched. Google and Google Scholar were used to identify other policy documents and relevant grey literature. Leading experts were consulted by email.

**Eligibility criteria for selected studies** Policies, policy guidance, strategies, codes of conduct, national service frameworks, national practice guidance, white and green papers, and reviews of policy—concerned with indicated suicide prevention approaches for children up to 18 years old. Limited by English language and published after 2000.

**Data extraction and synthesis** Data were extracted using a predetermined template. Second reviewers independently extracted 25%. Documents were categorised as international guidance, national policy and national guidance, and presented in a table providing a brief description of the policy, alongside how it specifically addresses suicidal CYP. Findings were further expressed using narrative synthesis.

**Results** 35 policy documents were included in the review. Although many recognise CYP as being a high-risk or priority population, most do not explicitly address suicidal CYP. In general, national guidance documents were found to convey that suicidal children should be assessed by a child and adolescent mental health practitioner but offer no clear recommendations beyond this.

**Conclusion** The lack of specific reference within policy documents to the treatment and care of needs of children who are suicidal highlights a potential gap in policy that could lead to the needs of suicidal children being overlooked, and varying interpretations of appropriate responses and service provision.

## BACKGROUND

Suicide is a global health policy priority, with nearly 800 000 lives lost to suicide annually. Suicide is arguably preventable. Reducing suicide rates is a target of WHO Mental Health Action Plan 2013–2020[1] in

which member states agreed to work towards reducing suicide rates by 10% by 2020 Globally, suicide prevention strategies have been established in 28 countries to date.[2]

### Strengths and limitations of this study

► This is the first scoping review to investigate how policy addresses suicidality in children and young people and provides new knowledge.

► The review authors employed a rigorous and methodological approach to the identification, screening and inclusion of policy documents, and following a clearly defined protocol.

► There is a lack of clear guidance in relation to how to conduct a systematic review of policy documents, which added complexity to the review process and reporting.

► The lack of specific reference to suicidal children within policy documents made it difficult to identify relevant documents, and there may be some we were unable to identify for inclusion.

International and government policies establish the context for the direction of resources for the development and delivery of services. Health policy provides a future vision (internationally, nationally or regionally), sets priorities and can include an action plan to achieve specific health-related objectives.[3] Public policies reflect international or national commitment and ambitions to address specific issues, but can vary in whether, and how they translate or relate to practice and whether there is a mandate for action.

Suicide is a leading cause of death of children and young people (CYP) worldwide, second only to accidental death.[4] It is estimated that as many as one in three children in some countries have considered suicide in the past year.[5] Although globally low/middle-income countries are often identified as having the highest rates of suicide overall, more economically developed countries report some of the highest rates of

suicide among [6] [7] CYP and in some it is the main cause of death.[8]

It is widely acknowledged that CYP have different needs to adults. A retrospective review of suicides among CYP found that there were even differences between the presenting issues for children under 15 years compared with adolescents (age ranges of adolescents was unspecified by the authors).[9]

Many countries commonly address the health needs of CYP separately to adults, with discrete policy and service provision, although definitions of what age range constitutes being a child, adolescent or young person varies greatly.[10] However, with most suicide prevention and mental health strategies now taking a universal approach, there is the potential for the specific needs of CYP to become lost.

Fortune and Clarkson[11] highlight the gulf that can often exist between suicide prevention policy and practice. They argue that although policy documents in New Zealand state that everyone who is suicidal should be assessed by a trained mental health professional, services are not adequately resourced to meet the demand. This is not unique to New Zealand, or to suicide prevention policy. The overall political context and policy agenda needs to be analysed in greater depth in order to make sense of the meaning conveyed within policy documents and attributed to them.[12]

Little is known about how policy addresses suicidality in CYP. Preliminary searches of review databases (Cochrane, DARE, JBI and the Campbell Collection) found that there had not been a review of worldwide policy in relation to CYP who are suicidal. Reviews to date have focused on the effectiveness of prevention and intervention strategies.[13] [14] Although generating valuable knowledge on the evidence base for interventions, they do not consider the policies behind such strategies and how this shape their focus and direction.

Mapping key policy documents worldwide and identifying how they address the treatment and care needs of suicidal CYP will: highlight the best practice for how policy can influence the resourcing of services; change practice and identify any gaps in policy provision for this vulnerable population. This knowledge will support countries who wish to develop new policies or further develop existing policies that address suicidality in CYP. The review question, objectives, search strategy and inclusion criteria were specified in advance and documented in a published protocol.[15]

### Objective
To map key policy documents worldwide and establish how they address the treatment and care needs of CYP who are suicidal.

### Inclusion criteria
#### Population
The key characteristics of the population were age and suicidality, neither of which have agreed universal definitions. As mentioned in the introduction, the authors recognise the disparity between the needs of children and young adults. Youth suicide research publications often tend to focus on older adolescents and young adults. While appreciating the complex challenges, including transition from child to adult services faced by 16–25 years old, this review informs a larger study, concerned specifically with a Scottish school-aged population. On this basis it was agreed to adopt a definition of a child as anyone under the age of 18 years, in line with: The Children (Scotland) Act 1995,[16] and the United Nations Convention of the Rights of the Child, 1998.[17] Policies solely about populations aged over 18 years were excluded, however, policies that cover the lifespan were included.

Suicidal behaviour is defined as acts of self-harm that result in death, as well as those with a non-fatal outcome. Non-suicidal self-injury is a term used to describe self-harming behaviour where there is no intent to die. This is most commonly used in the USA and became a discrete diagnostic category in the DSM V (Diagnostic and Statistacal Manual-5).[18] However, in the UK and some other European countries, definitions of self-harm are now also often used to include all non-fatal self-harming behaviours regardless of intent, and can include behaviour that may also be described as attempted suicide.[19] As this review is concerned with identifying relevant policies for children who are suicidal (had attempted to end their own life or were thinking about suicide), it was agreed not to use the term self-harm as a search term.

### Concept
Suicide prevention activities can be divided into three domains: universal activities which are aimed at everybody, including public health education programmes; selected or targeted interventions that aim to reduce the risk among specific high-risk groups and indicated interventions that may include treatments and are aimed solely at individuals presenting with suicidal behaviour. This review is solely concerned with identifying policy in relation to indicated activities, aimed at children (under 18 years of age) who are suicidal.

### Context
Identified policy documents were assessed for direct relevance to Scotland and the UK, or relevant to the context and population of the UK. Policies relating to indigenous populations, such as the Sami populations in Norway, Sweden and Finland, were consequently excluded[20]; but generic policies in postindustrial nations with developed economies such as Australia and New Zealand were included.[21] [22]

### Types of sources
Suicide prevention, like much healthcare policy, does not sit within clearly defined and labelled singular policy documents. As well as national suicide prevention strategies, there are more generic mental health strategies

**Table 1** Search terms

| Concept | Keywords |
|---|---|
| Children and young people (5–18 years) | Child*; "young people"; youth; adolesc*; teen*; paediatric |
| Suicide | Suicide; suicidal; |
| Policy | Policy; Procedure; Guidance; Strategy |
| Limit search by: | English Language; Published after 2000. |

or frameworks, and national guidelines such as those published by the National Institute for Health and Care Excellence (NICE) in the UK, which may contain specific references to indicated intervention approaches. Mapping policy requires a recognition of the variety of formats in which relevant documents may be found. Local government agencies and organisations also have their own individual policies and procedures, however, these should reflect the national approach. It was agreed that for the purposes of this review, policy documents would include: policies, policy guidance, strategies, codes of conduct, national service frameworks, national practice guidance, and white and green papers.[23] Reviews of policy documents centred on children who are suicidal were also eligible for inclusion as they contribute to the development of what is known in this area.

Given that the review aimed to map the present policy context, and most strategies are updated within a 10-year period, it was agreed to exclude any policy document or review published prior to 2000. Only those available as English Language versions were included.

### Search strategy

Keywords to be used as search terms (table 1) were generated from the review question.[15] Preliminary searches assisted in the refinement of these terms, and the identification of the most appropriate databases, platforms and websites. These terms were then amended for each of the databases and the exact terms, including any MeSH terms and subject headings used recorded (online supplementary file 1).

Four databases (CINAHL; Medline; PsycINFO; The Cochrane Database of Systematic reviews) and the websites of the following key government, statutory and non-statutory agencies were searched, focusing on postindustrial nations with developed economies in order to identify those with most applicability to the UK, for example, WHO; UNICEF, UK Government; Scottish Government; ScotPHO; UK NICE; National Office of Suicide Prevention (Ireland); Ministry of Health NZ; Australian Government Website and the Mental Health Commission Canada. Google and Google Scholar were also used to identify other policy documents and any relevant grey literature. Leading experts in the field were consulted via email.

All results were screened by title and abstract or executive summary by LG, with MM and ED screening a sample of 20%. Policy documents and articles were all screened in full by LG, and another sample of 20% was independently screened by MM and ED for inclusion. Disagreements were resolved by discussion, with the third reviewer acting as mediator.

### Method of the report

This review employed scoping review methodology to systematically identify relevant key policy documents following a pre-established search strategy and published protocol.[15] Scoping review methodology and guidance first outlined by Arksey & O'Malley[24] and further developed by Levac,[25] and the Joanna Briggs Institute[26] were used to inform the methodological process. The scoping review method was chosen as it allows for the synthesis of different types of study design. Thus, lending itself to the incorporation of different policy document formats (policies, policy guidance, strategies, codes of conduct, national service frameworks, national practice guidance, and white and green papers)[23] as well as any relevant existing published policy reviews. Suicidology of CYP is a newly emerging, highly sensitive and complex area of research, therefore well suited to scoping review methods.[24] The review is reported in line with the new Preferred Reporting Items for Systematic Reviews and Meta-Analyses (PRISMA) extension for Scoping Reviews.[27] Patients and the public were not consulted as part of this scoping review as it was not appropriate or applicable.

There were 43 records retrieved from PsycINFO, 193 from CINAHL, 12 from Medline and 49 from the Cochrane database of systematic reviews. After removing duplicates, there was a total of 297 records to be screened by title and abstract. Separate Excel spreadsheets were set up to catalogue the lists of references from each of the databases. After screening these results by title and abstract (completed in full by LG with a second reviewer independently screening for validity), all eight records to be screened in full text were found on CINAHL, although two were also found in duplicate on PsycINFO. Reasons for rejection of records included wrong topic, not in English and published before 2000.

After screening the eight articles identified by searching the databases (cross validated by a second reviewer), only two met the inclusion criteria.[28 29] Five were rejected as they were not policy documents about CYP who were suicidal, and although one seemed relevant in its references to the New Zealand suicide prevention strategy,[11] it was neither a policy document, nor a review of policy.

Internet searching was an iterative process, using keywords to search worldwide government, statutory and non-statutory agencies websites, with 39 potentially relevant policy documents identified. Although it is common practice in systematic reviews to screen the references of included documents for other potentially relevant papers, this occurred intuitively throughout the identification of policy documents, with one referencing others

within a country. Policy documents were only included for screening if by their title and description they seemed potentially relevant.

WHO Mindbank database houses links to member states National Suicide Prevention Strategies, however, many were unavailable in English. WHO mental health policy and services representative was contacted to request contact details of policy authors or country-specific contacts to enquire about English language versions. From these enquiries, an English language brochure outlining the content of the Swedish Suicide Prevention Strategy (known to be innovative for its zero suicide target) was obtained but we were unable to access the full document.[30] Although it was recognised that not every worldwide policy relating to CYP who were suicidal could be sourced, it was important to try and include all Scottish and UK-wide relevant policies. A request to the Scottish Government asking them to detail policies that should be included in the review, identified one further policy that had not been considered,[31] and this together with a related practice guide[32] were included for screening.

Screening of the 42 full-text documents was completed in full by the first reviewer (LG), with second reviewers each reviewing five independently (ED and MM), meaning a total of 25% was cross-validated. A meeting was then held to discuss the policy screening process, and to agree decisions about inclusion and exclusion. A total of 32 policy documents met the inclusion criteria. Reasons for exclusion were: document did not relate to or mention child suicidality; were not transferable to the UK or Scottish setting; was a review of systematic reviews; a newer version of the document is now available (online supplementary file 2). Together with the three published miscellaneous reviews/reports,[33–35] there were a total of 35 documents identified to be included—shown in the PRISMA diagram below[36] below (figure 1).

### Patient and public involvement
No public were involved in this review.

### RESULTS
The 35 included policy documents, ranged from: international guidance provided by the United Nations and WHO[4 37–39]; national suicide prevention strategies[22 30 40–47]; mental health strategies[48–53] and frameworks[54–58]; to national practice guidelines detailing how CYP who are suicidal should be assessed and treated.[19 29 59–62] The organisation and classification of these documents are illustrated in figure 2; providing language with which to describe the policy landscape.

Data extraction was completed by LG using a predefined template (online supplementary file 3) to collate key information about each of the documents including its aims and objectives, and how it related to the review question. A second reviewer independently extracted data for a sample of 25%. All three reviewers then met to discuss the process and outcomes. There was some variation in

the verbatim content extracted. This was regarded as a reflection of—(1) the size of the policy documents, and (2) because there were so few direct references to suicidal children, other content that could be interpreted as applicable but did not specifically mention suicide was also extracted from some to give context. The data extracted from each of the included policy documents were then tabulated (online supplementary file 4), categorised first by policy type, and then alphabetically by the country.

### International guidance
The UN 2030 Agenda for sustainable development[37] details the goals and action plan that all countries in the United Nations have agreed to deliver. Although it does not specifically mention suicidal CYP, reducing mortality by non-communicable illness (Goal 3.4) means that reducing suicide, which as a leading cause of death, must be a priority.

WHO mental health action plan 2013–2020[1] set a target that all countries should work towards reducing suicide rates by 10% by 2020. It recommended that countries adopt a life-course approach to mental health, reflecting an understanding of the impact of key stages in people's lives on health outcomes across their life span.[63 64] It promotes that countries create national policies and strategies to tackle suicide prevention prioritising at risk groups including 'youth'. However, other than the identification of 'youth' as a priority group it does not provide any other guidance on how countries should address suicidal CYP specifically.

Two other included documents published by WHO,[4 39] although also identifying suicidal CYP as a priority group, similarly do not go beyond this in terms of how their needs should be addressed. WHO Mental Health Gap Action Programme intervention guide provides generic guidance relating to interventions for all persons aged 10 years and over who are suicidal, suggests suicide should be included within an assessment, and advises that if young people feel suicidal they should talk to someone they trust and return to mental health support services.[39] No rationale is provided as to why 10 years of age has been selected. It does not differentiate between the assessment and treatment approach for suicidal children and adults. Clarification on this point was sought from WHO, but no response was received.

### National policy
WHO recommends that countries should develop suicide prevention and mental health strategies.[4 38] Ten suicide prevention strategies were included in this review,[22 30 40–47] five national mental health strategies and a young person's friendly version of the Canadian mental health policy.[48–53] As recommended by WHO,[38] the suicide prevention strategies adopt a universal and life-course approach. They generally provide demographic background information on suicides within their country and establish why it is a priority area. The policy documents describe their government's approach to tackling

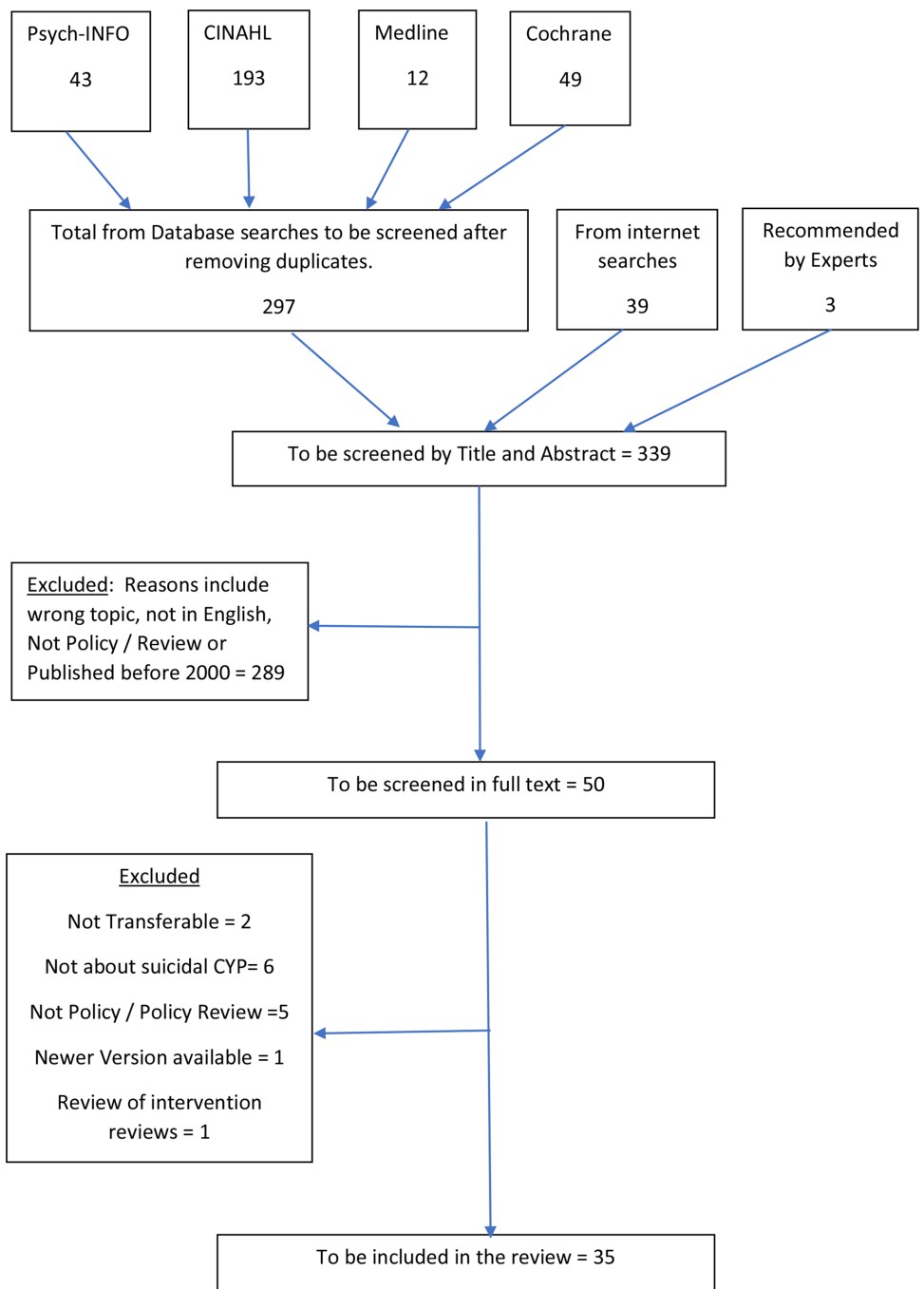

**Figure 1** Search results—PRISMA diagram. CYP, children and young people; PRISMA, Preferred Reporting Items for Systematic Reviews and Meta-Analyses.

suicide by detailing lists of aims, objectives and recommendations. Most strategies recognise that CYP are a priority group for universal and targeted suicide prevention activities (eg, universal whole school-based suicide prevention programmes and generic school counselling services), however, they do not differentiate between the indicated assessment and treatment offered to adults and that available to children who are suicidal. Some strategies made no reference to suicidal CYP,[22 40 46] including the Scottish Suicide Prevention Strategy 2013–2016.[46] Few mentions of therapeutic interventions specifically for CYP who are suicidal are made in the strategies.

The Irish strategy includes statements recommending that there should be early intervention, and 'enhanced support' available.[43] The New Zealand draft consultation document[44] contains suggestions that training teachers to talk to children who are suicidal, and having direct links between schools and psychologists will improve access to support for CYP who are suicidal. Notably New Zealand previously had a suicide prevention strategy directed specifically at 'youth' suicide,[65] but they have since adopted a universal policy covering the life course.[22] Similarly, other countries such as the USA previously had a strategy document that contained detailed objectives

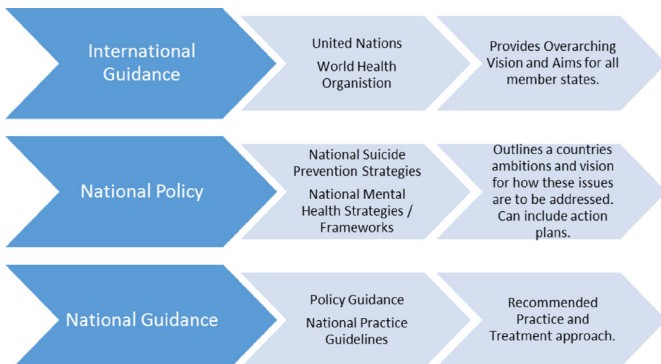

**Figure 2** Policy categories.

directly related to CYP (youth)[66] and has moved towards a much more generic approach.[45]

National mental health strategies were also found to take a life-course approach and were concerned with mental health promotion, supporting positive mental health and well-being and service delivery for those who experience mental ill health. The 10 national mental health strategies included in the review incorporate references to discrete service provision for CYP. Beyond generic school-based approaches to promoting positive mental health and well-being, there was also a focus on early intervention and easier access to child and adolescent mental health services.

Although most strategies refer to suicide as being a priority area in mental health provision; the Irish national mental health strategy[51] is the only one that specifically mentions suicidal CYP. Within a section dedicated to child and adolescent mental health under a heading 'Suicide and Deliberate Self-Harm',[51] it has recommendations that all children who present with self-harm should be assessed by the child and adolescent mental health team, and if appropriate receive treatment. It contains statements recognising adolescence as being a period of increased risk of suicide, and notably conveys that service provision should be the same across the whole country.

The Canadian Mental Health Strategy: A Youth Perspective[49] was produced to allow the document to be more accessible and relevant to young people. Although it does not provide specific recommendations in relation to the treatment and care of children who are suicidal, it urges that mental health services should be more accessible to all and highlights the Thunder Bay Youth Suicide Prevention Task Force as an example. This task force comprised 30 organisations working collectively to provide an immediate response.

The UK government policy 'No Health without mental health[50]' highlights the high incidence of self-harm among young people providing as an example that '10%–13% of 15–16 years old have self-harmed in their lifetime'. It includes a suggestion that all workers who are in contact with CYP should be aware of the issues surrounding this, and sets reducing the numbers of people of all ages who harm themselves as a target. However, the document does not contain a definition of self-harm in the glossary and

it is unclear if this suggestion includes those who attempt suicide. The policy contains no specific references to CYP who are suicidal.

The Scottish Government Mental Health Strategy 2017–2027[52] has a dedicated section to addressing the mental health needs of CYP. However, there is no reference to CYP who are suicidal. In relation to suicide, the strategy includes a statement that suicide prevention remains a government priority that will be dealt with separately (the Scottish Government's Suicide Prevention National Action Plan,[67] which was published following conclusion of the scoping review search—see the Discussion section).

Child and adolescent mental health services are delivered separately from adult services in many countries; consequently, there are distinct policies articulating a country's vision and aims for CYP's mental health. There are five such policy documents[54 56–58 68] included in this review, entitled as frameworks. The term 'frameworks' suggest they provide guidance for local authorities and those commissioning and delivering services; in the UK, it also denotes that they define standards of care. These frameworks include an outline of goals to prioritise and promote the mental health and well-being of CYP, and to deliver accessible services. None, however, specifically address CYP who are suicidal. Although containing recommendations for improving access to services and crisis support which could be applicable to CYP who are thinking about or have attempted suicide, this is not explicitly mentioned.

### National guidance

Another category of documents included in the review was national clinical guidance. These documents contain evidence-based recommendations for good practice. Although it is not compulsory to follow guidelines, organisations and clinicians must be aware of them and potentially justify their decision making should they choose to not implement them. Included clinical guidelines were: the UK NICE Guidelines for Self-Harm in over 8s: short term management and prevention of recurrence,[19] Self-Harm in over 8s long-term management and prevention of recurrence,[61] the New Zealand document: The Assessment and Management of People at Risk of Suicide,[60] the United States Preventative Task Force recommendations[29] and the American Academy of Paediatrics Guidance.[62] The Irish National Standard Operating Procedure for Child and Adolescent Mental Health Service (CAMHS)[69] was also included in this category, because it was a national document and specifically addressed the treatment and care needs of suicidal CYP. However, it was different to the other documents in this category because its implementation is compulsory.

The UK NICE guidelines for Self-Harm in over 8s: short-term management and prevention of recurrence,[19] apply to everyone over 8 years of age who presents following an incidence of self-harm, defining this as any act of self-harm regardless of intent. Therefore, these guidelines are applicable to anyone over 8 years who has attempted

suicide; although, it does not differentiate between the behaviours (with or without suicidal intent) in relation to treatment. They cover the immediate period following a presentation of self-harm (48 hours). The guidelines contain recommendations that all CYP who have self-harmed are admitted to hospital overnight in a paediatric ward (including adolescents aged over 14 years of age, if this is their preference) and they should be assessed by a specialist in child and adolescent mental health. This assessment should be the same as that for adults but also include a holistic assessment of their family situation, education. The only direct mention of suicide is that it is listed as a factor to be assessed. In the document that follows from NICE, Self-Harm in over 8s long-term management and prevention of recurrence,[61] the same definition of self-harm (to include self-harming behaviours with suicidal intent) is provided, and although suicidal intent is mentioned in relation to assessing risk, it also warns against using risk assessment tools to assess suicide risk. This reflects the lack of evidence for their effectiveness.[70 71] In the final recommendations section under 'Access to Services',[61] it states CYP who self-harm should be able to access all therapies and treatments available from Child and Adolescent Mental Health Services.

The New Zealand guidelines[60] are explicitly in relation to managing (all) people at risk of suicide. This document includes statements that all persons who are suicidal should be taken seriously and has a section dedicated to the treatment of children and adolescents. It includes recommendations that risk assessment of suicidal CYP should be conducted by someone trained in working with them, and that they should draw on information from the people around the child such as family and teachers as well as the child or young person themselves. In the background, information provided in the document it is stated that New Zealand has one of the highest rates of suicide among young people.

The USA document 'Screening for Suicide Risk in Adolescents, Adults and Older Adults in Primary Care: US Preventive Services Task Force Recommendation Statement[29],' includes a statement reflecting that there is no evidence to support any particular treatment or intervention for adolescents at risk of suicide, and not enough evidence to support assessment tools. The American Academy of Paediatrics, however, publishes very specific guidance for the treatment of adolescents presenting to primary healthcare following a suicide attempt or presenting with suicidal ideation.[28]

The National Operating Procedure for CAMHS in Ireland[69] was unique in that it specifically included standards of expected care and treatment for CYP who are suicidal across Ireland. It contained a statement that CAMHS would accept referrals for CYP where there are suicidal behaviours and intent. Similarly, to what was found within the other frameworks for child and adolescent mental health that were included in the review, this document[69] also included general statements about service provision that could be applied to CYP who are

suicidal, for example, references to CYP who need an immediate response, however, the term suicidal is not specifically used.

Beyond the policy documents included (international policies, national policies and national frameworks), there were very few other reports or reviews that were identified as relevant to the review question. Responding to Self-Harm in Scotland[34] is the report from the national self-harm working group and recognises that most people who self-harm do not intend to die. It includes a statement that young people are more likely to self-harm. One of its key recommendations is that there are clear referral pathways developed for people who self-harm, but it does not suggest what this might be, and is not specific to, or does not differentiate, between child and adult populations. A report commissioned in New Zealand to review the evidence on improving the outcomes for adolescents transitioning to adult services has a chapter dedicated to youth suicide,[35] which aims to provide an overview of the issue and prevention strategies. This chapter includes a description of how its national suicide prevention strategy and each of its goals applies to young people. The author concludes that the actions from the strategy can be applied to young people, although they are not specific to this population.

## DISCUSSION

This scoping review sought to answer the question: how does policy address the treatment and care needs of CYP who were suicidal? A total of 33 policy documents and 2 reports were included. However, overall, they offer little in relation to specific policy guidance for addressing suicidality in CYP. Suicide prevention strategies recognise that CYP are a priority population. However, the focus of these strategies is primarily on universal prevention approaches for CYP, such as whole school-based mental health and well-being education programmes or generic counselling services. Both national mental health strategies across the lifespan, and national frameworks for CYP's mental health, provide a blueprint for delivering services that are accessible to CYP who need them, when they need them. However, they do not specifically mention the population of children who are suicidal clearly enough to establish explicitly the care and treatment that they should be provided with. They also do not guarantee that the strategies or frameworks are delivered.

The national guidelines included within this review contain suggestions that CYP, who are self-harming or are suicidal, should be assessed by a child and adolescent mental health practitioner, and referred to CAMHS for treatment and therapeutic interventions. However, the included national frameworks for child and adolescent mental health barely reference CYP who are suicidal.

Recent research has found that even when there are national clinical guidelines recommending practice in relation to suicide intervention and treatment, clinical staff teams are not always aware of these and

implementation varies.[72] This strengthens the case for countries adopting a model, like that in Ireland, where implementing the Child and Adolescent Mental Health Services SOP[69] is compulsory, and goes beyond guidelines for recommended practice.

One of the reasons for the identified paucity of policy direction in providing interventions and treatments for CYP who are suicidal is perhaps the lack of evidence for the effectiveness of any particular treatment approach.[73 74]

It could also be argued that the function of policy is not to address the treatment and care needs of specific populations, but provide a future vision and action plan to achieve this, which can be interpreted and disseminated within a local context. However, the lack of dialogue around CYP who are suicidal within the documents reviewed highlights a gap in policy provision for this population. The review of the New Zealand Suicide Prevention Strategy[35] demonstrated that the generic goals set out in the strategy could be applicable to young people; however, this was not obviously apparent from the document itself. This may be true for other national suicide prevention strategies and national mental health strategies. However, by not being explicit about their relevance to CYP who are suicidal, it could mean that the needs of this population are overlooked by the local government agencies charged with interpreting, implementing and resourcing them. It may also lead to large variations in terms of service design and delivery across different local authorities.

## LIMITATIONS

This is the first scoping review to consider how policy addresses the needs of CYP who are suicidal, and provides unique insight into this policy domain. However, the lack of methodological guidance for conducting policy reviews made this challenging. While we recognise some of this study's limitations, we have tried to be explicit in our methodology and conduct the review with rigour. Additionally, the lack of any previous description of the suicidality policy landscape for CYP, made identifying and sourcing relevant documents complex. The systematic searching of primarily journal-based databases returned very few relevant documents. Searching government websites for terms such as: 'child', and 'young people', and 'suicide', was also problematic because many of the key documents include little direct references to CYP who are suicidal. The search for policy documents was more intuitive than anticipated, in part due to the paucity of research in this area. One of the key findings was that there is a gap in policy specifically addressing this population, but this gap also contributed to the difficulty in finding relevant policies to be included.

The identification of Ireland's standard operating procedure for CAMHS[69] suggests that there may be clear protocols for child and adolescent mental health services, and practitioners available in other countries. However, these documents tend to vary between organisations and local authorities/states and were excluded from this review because they were not national. Further exploration of these local policies, or purposive searching for other international CAMHS protocols should be considered within any future policy research in this area.

Although not a prerequisite in a scoping review, triangulating screening and data extraction helped to identify that there is little policy dialogue about indicated suicide prevention strategies for children. This lack of specific reference to the care needs and pathways for suicidal children meant that the documents were open to subjective interpretation. For example, although parts of policies could be interpreted as being applicable to this population of CYP, in attempting to extract verbatim the text that addressed them reviewers struggled to identify significant relevance.

As the review was limited to English language many of the suicide prevention policies had to be excluded, including those of the Nordic Nations who are known to have advanced mental health and suicide action plans, as they could not be translated. These countries may make their policies available in English in the future as they have with 'Plan for suicide prevention among the Sàmi people in Norway, Sweden and Finland[20]' and they could then be included in a future review.

The policy landscape is constantly changing and evolving. Two highly relevant documents were published following completion of the systematic literature search. The Scottish Government published an updated Suicide Prevention Strategy,[75] which contains acknowledgements that 'CYP require a specific focus'. Recommendations within the policy document itself remain largely at a universal prevention level, for example, training teachers. The strategy clearly includes a recommendation that all children should have access to crisis support when they need it, and that it is the governments' intention to 'transform' child and adolescent mental health services, having appointed a CYP's mental health task force. However, it also contains a statement suggesting that suicide rates in children are falling, which is contrary to reports from other sources which suggest that they are increasing,[76] and that rates in Scotland are higher than other parts of the UK.[8] (Recent changes in coding of deaths in line with ICD-10[77] (International Classification of Diseases - 10) (deaths with undetermined intent are now being recorded as suicide) has had implications on recent suicide rates. Additionally, the Scottish Government does not publish annual suicide rates in populations of children aged under 15 years. They provide a statement that this statistic could be misleadingly high for children in this 'extreme' age group as a higher proportion of deaths are recorded as undetermined.[78]

Another key document published latterly was the UK-wide Self-harm and Suicide Competence Framework CYP.[79] This document is intended to outline the key competencies required of professionals working with CYP who self-harm or are suicidal. Identifying that the knowledge and skills of those who support children who

self-harm or are suicidal requires different competencies to those who work with adults is undoubtedly a positive development. Within the document, it is emphasised that a person-centred approach should be taken towards CYP who have self-harmed or are suicidal, and they are treated with compassion and respect. It contains acknowledgement of the challenges in assessing suicide risk: scales and risk assessment tools have a low prediction value; and there remains a lack of evidence base for any effective interventions. However, it goes on to promote the use of dialectical behavioural therapy (DBT) and mentalisation behavioural therapy (MBT) as specific interventions for use by mental health professionals based on the identification of positive effect in single trials of DBT and MBT (79:27).This is then followed by a statement warning that the generalisability of these approaches is unknown.

Overall, the report conveys the complexity involved in understanding the needs of children who self-harm and are suicidal and is a welcome guide to practitioners and service providers, concerned with the supervision and training needs of their workforce. However, it remains within the realm of recommendations, its application is not compulsory, and it highlights the paucity of evidence to support effective treatment models for this vulnerable population.

## IMPLICATIONS FOR FUTURE RESEARCH

This review highlights the need for further research in several areas. It establishes a need for more robustly defined policy review methodology, as well as a deeper exploration of the potential gap in policy provision for suicidal CYP.

Although scoping review methodology lends itself well to policy review, ensuring that the search strategy and identification of policy documents is reliable is complicated by variation in document formats, and titles that do not describe the issue in focus. This presents similar issues to those found when trying to identify qualitative literature for the purposes of review synthesis,[80][81] and learning from developments in the field of qualitative evidence review could support development of more robust policy review methodology.

Application of discourse analysis or interpretative policy analysis[82] may help to understand the meaning of the policy dialogue, as policy can in and of itself support the construction of or denial of social issues.[83] Consideration of how the problem of childhood suicidality is represented in policy documents could provide valuable insight[84] into the politics of addressing this highly sensitive subject, and the needs of these children.

Additionally, widening the inclusion criteria in future reviews to include more local policies would provide further knowledge on how national policy is interpreted and applied at a local level. Exploring whether there are variations in interpretation locally, and if in fact the specific needs of suicidal CYP do get lost in translation is a knowledge gap that needs addressed.

Policy documents need to be written in such a way that they are careful not to exclude people and are therefore often very generic. Taking a lifespan approach to mental health policy and suicide prevention strategies supports the holistic understanding that mental health is not just about the absence of illness. Mental ill health, periods of distress and suicidality are all fluid concepts that can touch all our lives at different points. The aim of these strategies is for governments to explicate their commitment to addressing these issues, and to supporting people of all ages who are affected by them. However, this review suggests that by not specifically naming suicidal CYP as a group that should have immediate access to services or supports, and what this might look like, there lies a danger that generic policy statements are too open to interpretation. This could have implications for the local funding, commissioning and delivery of child and adolescent mental health services. Policy-makers should clarify their ambitions for how the treatment and care needs of suicidal CYP should be addressed in future policy documents.

This review provides practitioners with an overview of the international and national policy context within which they work, informing their practice and providing key knowledge. It may support their understanding of practice guidelines in relation to CYP who are suicidal and equip them with a reference resource from which to draw on.

## CONCLUSION

This scoping review mapped key policy documents worldwide and established how they addressed the treatment and care needs of CYP who are suicidal. Categorising these documents by International Policy, National Policy and National Guidance revealed that despite the assertion that CYP are a priority target population within policy documents, their content mainly promotes the use of universal prevention strategies and does not specifically address the treatment and care of CYP who are suicidal. This highlights a potential gap in policy that could lead to the needs of this very vulnerable group being overlooked, and varying interpretations of how they should be provided for. National guidelines (in the UK, and New Zealand)[19][60] and Ireland's SOP for CAMHS[59] contain recommendations that CYP who are considered to be at risk of suicide are assessed by a child and adolescent mental health practitioner, however, stop short of recommending treatments and interventions beyond this.

**Contributors** LG led this review and paper, ED and MM contributed to the development of the protocol. LG ran the searches and applied the selection criteria. MM and ED verified the selection of documents, independently screening all identified documents by title and abstract, and 50% in full text. LG completed data extraction, with MM and ED independently extracting data from 25% of included documents. LG mapped the included documents and wrote the first draft of the paper. This was reviewed and commented on by MM and ED. All authors read and approved the final manuscript.

 

**Funding**  LG is an ESRC funded PhD student.

**Competing interests**  None declared.

**Patient consent for publication**  Not required.

**Provenance and peer review**  Not commissioned; externally peer reviewed.

**Data availability statement**  All data relevant to the study are included in the article or uploaded as supplementary information. Online supplementary tables are provided detailing MeSH terms, reasons for exclusion, the data extraction template used and the table of included policies.

**ORCID iDs**
Lynne Gilmour http://orcid.org/0000-0001-8876-5590
Edward Duncan http://orcid.org/0000-0002-3400-905X

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
