## [Reviewer comments · BMJ Open]

ARTICLE DETAILS

TITLE (PROVISIONAL)	Policy addressing suicidality in children and young people: an international scoping review.
AUTHORS	Gilmour, Lynne; Maxwell, Margaret; Duncan, Edward

VERSION 1 – REVIEW

REVIEWER	Charlotte Connor University of Warwick, England
REVIEW RETURNED	17-Apr-2019

GENERAL COMMENTS	A very timely and useful review with significant implications for the treatment of children and young people. The age range limit was 18 years, however, and whilst I appreciate that the authors used The Children (Scotland) and UNICRC definitions of a child, I would be interested in seeing a brief discussion of this cut-off point, given that in the UK, for example, CAMHS services are moving towards a 0-25 year model. This should include the difficulties encountered in treatment for young people who fall between childhood and adult services.
---

REVIEWER	Elizabeth Suarez Soto Universitat de Barcelona, Spain
REVIEW RETURNED	22-May-2019

GENERAL COMMENTS	This paper presents a research about "Policy addressing suicidality in children and young people: an international scoping review." The content is extremely important and it has very low visibility and very scarce research in scientific literature. Therefore, the manuscript is innovative and relevant. On the other hand, the paper is clearly written and well organized, using appropriate statistical analysis. Results are interesting and relevant for policy makers and practitioners. In general terms, the sample is appropriately selected and described, and results and tables are appropriately presente.
--

REVIEWER	Miharu Nakanishi Tokyo Metropolitan Institute of Medical Science, Japan
REVIEW RETURNED	01-Aug-2019

GENERAL COMMENTS	The present study collected policy documents to examine presence of specific initiatives for children and young people who are suicidal. It would be questioned about design and methodology adopted in the study. 1. Collection of documents: the present study seems to collect
--

	documents available in English only. It might be less common approaches, as some researches in dementia (Durepos et al. J Pain Symptom Manage 2017) or schizophrenia (Gaebel et al. Br J Psychiatry 2005) collected non-English documents as well. 2. Collection of countries: rationales of country selection are critical in the study. Although the authors used academic search engines, the study was focused on policy issues that are not necessarily reported in academic reports. As is mentioned in BACKGROUND, WHO has a list of countries which adopted suicide prevention strategies. A strong explanation should be provided on why not start with collecting the documents based on the list. 3. Inclusion and exclusion criteria: the present study included mental health policies in the analysis. It would be questioned, because the WHO report (2018) defines a specific role of suicide prevention policies in suicide reduction that could not be duplicated by other policies. 4. Reference, recommendations or guidelines: in the manuscript, it is not defined on how and what types of treatment and approach to meet care needs should be identified for children and young people in policy documents. Are there any references, international recommendations or guidelines available?
--	--

VERSION 1 – AUTHOR RESPONSE

Reviewer 1 Comments:	Authors Response
A very timely and useful review with significant implications for the treatment of children and young people. The age range limit was 18 years, however, and whilst I appreciate that the authors used The Children (Scotland) and UNICRC definitions of a child, I would be interested in seeing a brief discussion of this cut-off point, given that in the UK, for example, CAMHS services are moving towards a 0-25 year model. This should include the difficulties encountered in treatment for young people who fall between childhood and adult services.	We would like to thank you for your comments about our review. We welcome your feedback regarding the difficulties experienced by older adolescents and young adults. In our introduction we refer to the differences between the needs of children and older young people (page 3). We have amended the narrative on page 5 in relation the sample population to read “The key characteristics of the population were age and suicidality, neither of which have agreed universal definitions. As mentioned in the introduction the authors recognise the disparity between the needs of children and young adults. Youth suicide research publications often tend to focus on older adolescents and young adults. Whilst appreciating the complex challenges, including transition from adult / child services faced by 16-25 year olds, this review was conducted in Scotland and informs a larger study taking place here, concerned specifically with a school aged population. On this basis it was agreed to adopt a definition of a child as anyone under the age of 18 years, in line with: The Children (Scotland) Act 1995 (16), and the UNCRC (United Nations

	Convention of the Rights of the Child, 1998) (17). Policies solely about populations aged over 18 years were excluded, however policies that cover the lifespan were included.”
Reviewer 2 Comments:	Authors Response
This paper presents a research about "Policy addressing suicidality in children and young people: an international scoping review." The content is extremely important and it has very low visibility and very scarce research in scientific literature. Therefore, the manuscript is innovative and relevant. On the other hand, the paper is clearly written and well organized, using appropriate statistical analysis. Results are interesting and relevant for policy makers and practitioners. In general terms, the sample is appropriately selected and described, and results and tables are appropriately presented.	We would like to thank you for your comments about our review. They are much appreciated.
Reviewer 3 Comments:	Authors Response
The present study collected policy documents to examine presence of specific initiatives for children and young people who are suicidal. It would be questioned about design and methodology adopted in the study.	We would like to thank you for reading and commenting on our review.
Collection of documents: the present study seems to collect documents available in English only. It might be less common approaches, as some researches in dementia (Durepos et al. J Pain Symptom Manage 2017) or schizophrenia (Gaebel et al. Br J Psychiatry 2005) collected non-English documents as well.	We would like to thank you for this comment and agree that it is a limitation of this study that we were only able to include policies that were written in English. We refer to this in our limitations section “As the review was limited to English language many of the suicide prevention policies had to be excluded, including those of the Nordic Nations who are known to have advanced mental health and suicide action plans, as they could not be translated. These countries may make their policies available in English in the future as they have with “PLAN FOR SUICIDE PREVENTION AMONG THE SÁMI PEOPLE IN NORWAY, SWEDEN, AND FINLAND” (20) and they could then be included in any future review.” (Page 20)

Collection of countries: rationales of country selection are critical in the study. Although the authors used academic search engines, the study was focused on policy issues that are not necessarily reported in academic reports. As is mentioned in BACKGROUND, WHO has a list of countries which adopted suicide prevention strategies. A strong explanation should be provided on why not start with collecting the documents based on the list.	We appreciate your comment and the opportunity to clarify further in the text that we did search the WHO Mindbank database, and contacted them directly to request support in obtaining the most relevant policies pertaining to our review that were available in English. We have amended the text on page 9 to now read. “The World Health Organisation (WHO) Mindbank database houses links to member states National Suicide Prevention Strategies, however, many were unavailable in English. The WHO mental health policy and services representative was contacted to request contact details of policy authors or country specific contacts to enquire about English language versions. From these enquiries an English language brochure outlining the content of the Swedish Suicide Prevention Strategy (known to be innovative for its zero suicide target) was obtained but we were unable to access the full document (30).” We also express within our description of context that we wished only to include documents which could be considered transferable to a UK setting. “Identified policy documents were assessed for direct relevance to Scotland and the UK, or relevant to the context and population of the UK. Policies relating to indigenous populations such as the Sami populations in Norway, Sweden and Finland, were consequently excluded (20); but generic policies in post-industrial nations with developed economies such as Australia, and New Zealand were included (21,22).”
Inclusion and exclusion criteria: the present study included mental health policies in the analysis. It would be questioned, because the WHO report (2018) defines a specific role of suicide prevention policies in suicide reduction that could not be duplicated by other policies.	We appreciate your comment and have clarified more explicitly our selection of sources on page 6 “Suicide prevention, like much health care policy does not sit within clearly defined and labelled singular policy documents. As well as national suicide prevention strategies, there are more generic mental health strategies or frameworks, and national guidelines such as those published by the National Institute of Clinical Excellence (NICE) in the UK, which may contain specific references to indicated intervention approaches. Mapping policy requires a recognition of the variety of formats in which relevant documents may be found. Local government agencies and organisations also have

	their own individual policies and procedures, however these should reflect the national approach. It was agreed that for the purposes of this review, policy documents would include: policies, policy guidance, strategies, codes of conduct, national service frameworks, national practice guidance, and white and green papers (23). Reviews of policy documents centred on children who are suicidal were also eligible for inclusion as they contribute to the development of what is known in this area.”
Reference, recommendations or guidelines: in the manuscript, it is not defined on how and what types of treatment and approach to meet care needs should be identified for children and young people in policy documents. Are there any references, international recommendations or guidelines available?	Thank you for this comment. This review attempted to identify whether there were any policy recommendations or guidelines available specifically in relation to treatment approaches for this population. In the discussion section we note that there is no agreed treatment approach for children and young people who are suicidal. “One of the reasons for the identified paucity of policy direction in providing interventions and treatments for children and young people who are suicidal, is perhaps the lack of evidence for the effectiveness of any particular treatment approach (73,74).” (Page 18) As was highlighted in the limitations section there was a recent document published following the completion of the scoping review – “the UK wide Self-harm and Suicide Competence Framework children and young people (79)” (Page 21)